# Preparation of Phage Display cDNA Libraries for Identifying Immunogenic Tumor Antigens: Challenges in Functional cDNA Presentation and Approaches to Overcoming Them

**DOI:** 10.3390/v16121855

**Published:** 2024-11-29

**Authors:** Nuša Brišar, Katja Šuster, Andrej Cör

**Affiliations:** 1Faculty of Health Sciences, University of Primorska, 6310 Izola, Slovenia; 2Faculty of Medicine, University of Ljubljana, 1000 Ljubljana, Slovenia; 3Valdoltra Orthopaedic Hospital, 6280 Ankaran, Slovenia; katja.suster@ob-valdoltra.si (K.Š.); andrej.coer@ob-valdoltra.si (A.C.); 4Faculty of Education, University of Primorska, 6310 Izola, Slovenia

**Keywords:** nanoparticles, phage display technology, filamentous bacteriophages, cDNA libraries, tumor antigens, biopanning, immunotherapy

## Abstract

Cancer continues to represent a substantial burden in terms of its morbidity and mortality, underscoring the imperative for the development of novel and efficacious treatment modalities. Recent advances in cancer immunotherapy have highlighted the importance of identifying tumour-specific antigens, which can assist the immune system in targeting malignant cells effectively. Phage display technology has emerged as an effective tool for the discovery of novel antigens through cDNA library screening, representing a significant advancement in the field of immunological research. This review examines the discovery of tumour antigens using phage display technology, emphasising the construction of cDNA libraries, their subsequent display on bacteriophages and the utilisation of diverse biopanning techniques. These elements play a pivotal role in advancing the discovery of novel tumour antigens and the development of targeted cancer therapies. This review addresses the challenges associated with the filamentous phage display of cDNA libraries and proposes strategies to improve the effectiveness of this approach, encouraging further research for clinical applications.

## 1. Introduction

Despite remarkable medical advancements, cancer remains the second leading cause of morbidity and mortality worldwide. For years, the cornerstones of cancer treatment have been surgery, chemotherapy and radiation therapy [1]. However, continued progress in the understanding of the biology and molecular background of several types of malignancies has resulted in the development of advanced cancer immunotherapies that harness the natural mechanisms of the immune system to recognise and eliminate malignant cells [2].

When normal cells undergo malignant transformation, they acquire tumorigenic genetic alterations that cause changes in the protein structure or the timing or quantity of protein expression [3]. The immune system can often recognise these altered tumour proteins (i.e., antigens) as non-self-proteins and subsequently kill tumour cells. Recognition by effector T lymphocytes, crucial cells in the cancer immunity cycle, is possible only when these antigens are presented on the surface of human leukocyte antigen (HLA) molecules [4]. Therefore, systematic identification of tumour antigens that can be incorporated into treatments that induce T-cell-mediated tumour rejection is indispensable to rapidly advancing the development of several cancer immunotherapies [5].

The discovery and characterisation of new antigens are achievable using a diverse range of high-throughput multi-omics analyses, including tandem mass spectrometry (MS/MS) and next-generation sequencing (NGS) or a combination of both [6,7]. An indirect method for antigen discovery is also provided by screening cDNA expression libraries (e.g., using sera from cancer patients). Different surface display technologies [8] enable screening cDNA libraries for peptides that interact with a bait protein and thus the detection of interaction partners with specific desired properties. For this purpose, bacteriophage surface display is by far the most widespread utility [9,10,11,12].

Bacteriophages, or phages, are viruses that specifically infect bacteria and subordinate their cellular machinery to create new phage particles, ultimately resulting in the release of new phage virions [13]. George Smith paved the way for bacteriophage engineering in 1985 when he demonstrated the fusion of (poly)peptides to the outer proteins (i.e., capsid) of phages, enabling surface display [14]. His work, for which he was awarded the Nobel Prize in 2018, laid the groundwork for the development of phage display technology and biopanning methods [15].

Since then, phages have been exploited in many different research fields, among these also their exploitation as target nanocarriers in cancer immunotherapy. In this way, a favourable presentation of tumour antigens to immune cells can be achieved to direct them to act against tumour cells [16]. Lately, there has been an increase in the research on screening the cDNA expression libraries displayed on the surface of filamentous phages. Cloning a wide repertoire of cDNA libraries prepared from tumour lysates into phages and further selection of (poly)peptides using in situ, in vitro, in vivo and ex vivo screening methods enable the identification of new cancer-specific ligands that could potentially be used as therapeutic targets or biomarkers [17].

Despite the latest developments in the field, there are still some limiting challenges in the filamentous phage display of cDNA libraries. Apart from being subjected to the intrinsic biological limitations in phage and bacterial life cycles [18,19,20], the synthesis of fusion proteins is frequently prevented by translational stop codons in reverse-transcribed eukaryotic mRNA [18], shifts in reading frames or the misorientation of inserted fragments [21]. However, several strategies exist and can be successfully adopted to overcome these challenges and make cDNA libraries more accessible for phage display.

This review aims to describe tumour antigen discovery, focusing on recent developments in the use of antigen cDNA libraries and phage display. During this review, we critically expose and discuss key challenges and limitations in the field of filamentous phage display of cDNA libraries and the currently employed solutions and suggest new possible forms of improvement. Outlining the potential of this technique for use in cancer immunotherapy, we expect this review to open new ideas and possibilities for new research regarding advanced cancer treatment.

## 2. Filamentous Bacteriophages and Phage Display Technology: An Overview

Bacteriophages (or phages for short) are a class of prokaryotic viruses abundant in nature and harmless to humans. Although the families of phages greatly vary in their shapes, sizes and life cycles, they uniformly consist of nucleic acids and proteins. The phage genome can come in the form of single-stranded (ss) or double-stranded (ds) DNA or RNA and is encapsulated by a protein capsid [13]. There are two types of bacteriophages, depending on their mode of replication: virulent (lytic) and lysogenic (lysogenic). Lytic bacteriophages subordinate the bacterial cellular machinery for the replication of bacteriophage genetic material and the synthesis of bacteriophage proteins, resulting in the destruction (lysis) of the bacterium and the release of thousands of new identical bacteriophage virions. Lysogenic bacteriophages incorporate their genome into the bacterial genome, producing a prophage that replicates along with the bacterial DNA as the bacterium replicates. However, a lysogenic bacteriophage can also enter the lytic cycle and start to build new virions [22]. Phages were independently discovered by Frederick Twort in 1915 and Félix D’Herelle in 1917 for their use against bacterial infections; however, research on this theme was abandoned upon the discovery of antibiotics. It was only in 1985 that bacteriophage research for human therapy came to the fore again [23]. In fact, with phage display technology, phages began to be used (a) as a platform for displaying tumour antigens for cancer immunotherapy or (b) to uncover novel tumour antigens for the design of new delivery systems [24].

In phage display technology, phages can be engineered to present foreign peptides or proteins as a fusion with phage coat proteins. The power of phage display technology is in establishing the link between the genotype (the gene sequence encoding the displayed peptide/protein) and the phenotype (the displayed peptide/protein). A gene sequence encoding a peptide or a protein of interest is inserted into a coat protein gene. Through the inherent phage machinery, a recombinant fusion protein is formed and subsequently incorporated into the phage capsid and displayed on the surface of the phage [13,23,24,25]. Commonly, phage display employs either phage or phagemid vectors.

Among these phage vectors are the most widely and successfully exploited filamentous phages of the *Inoviridae* family M13, fd and f1 [15]. They are peculiarities in the world of phages in terms of their appearance and mode of assembly, as they do not match the archetypal phage head and tail structure and do not lyse bacterial hosts in their life cycles [20]. In the case of M13 phages, N-terminal fusion of exogenous (poly)peptides to pIII and pVIII and C-terminal fusion to pVI coat proteins proved to be the most effective for the display of cDNA libraries. These differ in the number of (poly)peptides with which the capsid is decorated. PVIII is present in 2700 copies surrounding the whole phage and is preferred for the display of short peptides, while pIII and pVI are expressed in 5 or 4 copies at one end of the phage and are used for the display of larger peptides [26]. Sometimes, for example, if the size of an insert is very large, such a display may cause slower phage assembly and disrupt the arrangement of the pIII proteins, leading to reduced phage infectivity. In such cases, phagemid systems are preferred, which combine selected genomic features of phages and bacterial plasmids, providing several advantages [27]. Phagemids are phage-derived vectors that do not encode for all of the functional and structural proteins of the phage, instead carrying only the necessary origins of replication, a cloning cassette, a selective marker and one type of coat protein that is employed for the display [15]. They contain the plasmid origin of replication allowing for circular dsDNA replication, the filamentous phage origin of replication for allowing ssDNA replication and the phage-packing signal, which allows for the packing of ssDNA into the phage capsids. Accordingly, the phagemid can maintain itself as a plasmid in a bacterial host but is not able to finish the assembly of progeny phage virions independently. Only when bacteria harbours both the phagemid and the helper phage, which contains the genetic elements for the remaining phage coat proteins that are missing in the phagemid, can infective particles with the same morphology as the filamentous phages be produced and released from infected bacteria [27].

## 3. Discovery of Tumor Antigens Using Phage Display Technology

### 3.1. Construction of cDNA Libraries and Their Display on Bacteriophages

The preparation of phage-displayed cDNA library begins to take place with the construction of a cDNA library encoding different peptides/proteins. The first step is the isolation of cytoplasmic RNA from the studied cell or tissue type. Cytoplasmic RNA consists primarily of ribosomal RNA (rRNA), transfer RNA (tRNA) and, to a lesser extent, messenger RNA (mRNA). Further separation of eukaryotic mRNA from the RNA mixture using affinity chromatography is possible due to its unique structure. The 3′ end of mRNA rich in adenylate residues is referred to as the poly-adenosine (poly-A) tail. With it, the mRNA binds to latex or magnetic particles pre-conjugated with chains of thymidylate (oligo-dT), which is complementary to the poly-A tail. As the cell extract passes through the purification column, only the mRNA base-pairs with oligo-dT; on the contrary, the tRNA and rRNA and other unbound molecules are washed away [28,29]. The resulting extract of total cellular mRNA contains many different mRNA molecules encoding different proteins [29] (Figure 1).

In the next step, an enzyme reverse transcriptase synthesises cDNAs corresponding to all mRNAs, resulting in mRNA-cDNA duplexes. At the 3′ end of the mRNA, the bound oligo-dT serves as a primer for the enzyme which extends the 3′ end of the mRNA [29]. The yield of cDNA containing the entire coding sequence depends on the quality of the mRNA and the presence of secondary mRNA structures, which reduce the efficiency of the reaction [28]. When the mRNAs in mRNA-cDNA duplexes are removed, the remaining cDNAs are converted into double-stranded cDNAs by the enzymes terminal transferase and DNA polymerase. The enzyme terminal transferase, which does not need a template for its function, adds several residues of a selected nucleotide (e.g., guanine) to the 3′ end of the cDNA. A synthetic oligo-dC fragment is then complementarily bound to the 3′ oligo-dG and serves as a primer for the enzyme DNA polymerase, which synthesises DNA strands complementary to the original cDNAs. Each double-stranded cDNA fragment corresponds to each of the mRNAs in the cell extract [29] (Figure 1).

Restriction-cleaved double-stranded cDNA fragments are further cloned into various phage or phagemid vectors. Because cDNA represents a complementary sequence of mRNA, it includes only the exon portions of the gene. Consequently, the amino acid sequence of a peptide/protein displayed on a phage can be determined directly from the nucleotide sequence of its corresponding cDNA variant within the phage genome [29] (Figure 1).

A promising strategy in cancer therapy is the use of tumour-targeting peptides to selectively deliver drugs or active agents to solid tumours. Several tumour-homing peptides have been identified through combinatorial library screening, including glioma-targeting peptides, tumour-vasculature-targeting peptides and peptides targeting tumour invasion, as well as cell-penetrating and tumour-penetrating peptides [30,31,32]. In particular, phage display libraries have facilitated the discovery of novel ligands for cancers such as gastric cancer [33], melanoma [34] and breast cancer [35,36]. This approach has also contributed to the development of numerous phage-derived antibodies, many of which have progressed to clinical trials, leading to the approval of over a dozen therapeutic antibodies [37].

### 3.2. Affinity Selection Strategies for Screening Bacteriophage Display Libraries

The surface of a phage can be variably decorated with multiple copies of the polypeptide of interest, and within a phage population, a multiplicity of unique motifs are rapidly and easily displayed to create a phage-displayed library [23]. Through an affinity-driven process, commonly known as biopanning, the phage-displayed libraries are screened against a wide variety of targets, and the target-specific (poly)peptides displayed on the phage (e.g., new cancer ligands) are subsequently identified. In biopanning, the phage display library is first incubated with the target for a defined period of time. This is followed by a series of stringent washing steps. Phages with binding affinity to the desired target molecule (e.g., the bait protein) are retained, while weakly bound or unbound phages are washed away. The interacting phages are then eluted and amplified by re-infecting bacteria, thus creating a new phage library with higher-affinity and specific binders used for a subsequent three to five rounds of panning. Finally, the peptides/proteins with the highest affinity are identified through DNA sequencing of the selected phages [15,24,38] (Figure 2). Applying phage display biopanning techniques enables high-throughput, rapid, cost-effective screening and the selection of phage-displayed peptides/proteins targeting overexpressed tumour cell receptors, as well as components in tumour microenvironment like immune cells exerting pro-tumour activities, tumour vasculature and endothelial cells [17]. These features have to be evaluated when choosing a platform for new tumour antigen discovery and cancer vaccine design.

The biopanning procedure can be performed using various approaches. One common method, in situ screening, involves immobilising the purified target molecule onto a solid support [17,38]. Despite this being the most straightforward and frequently used method, there are still some shortcomings that need to be considered. When a target molecule is artificially attached to a carrier, its secondary structure may be misrepresented, posing the risk of isolating a phage clone unable to bind or with non-specific binding upon exposure to the in vitro or in vivo system. Biopanning can also be carried out in vitro on cell lines to identify cell-surface-interacting peptides. This approach preserves the biological functions of cells, the proper folding and 3D structure of the proteins, the level of receptor expression and their interaction with other proteins [17]. The utilisation of both in vitro and in vivo biopanning approaches in phage display technology has facilitated the discovery of peptides with high specificity and affinity which could be employed as therapeutics in drug delivery, diagnostics and tumour targeting [39]. Ex vivo screening in an isolated sample enables the selection of certain rare cells in a heterogeneous population, and in vivo screening in animals enables the identification of organ- or disease-specific homing peptides [17,38]. Yet the risk is that the selected peptides may not be useful for human treatment due to differences in the peptide binding between different species [17]. The half-life of administered phages is also questionable [40]. To overcome these hindrances, screening libraries in vivo in patients in the terminal stage of cancer allows for better mimicry of natural heterogeneous cells or the conditions of the human body [41] (Figure 2).

## 4. Phage Display of cDNA Libraries: Challenges in Functional cDNA Presentation and Employed Solutions

Phage coat proteins are synthetised in the controlled and reducing environment of the bacterial cytoplasm and are translocated across the inner membrane into the oxidising environment of the periplasm. Minor coat proteins pVI, pVII and pIX are synthesised without signal peptides, whereas pIII and major protein pVIII are synthesised with an amino terminal signal peptide which directs their insertion into the cytoplasmic membrane, where they reside before being assembled into a new phage virion. Any peptide/protein fused to the periplasmic portion of these phage proteins theoretically should be likely to be packed into a phage virion, provided it can be translocated efficiently into the membrane and fold properly, escape degradation in the periplasm and not interfere with the process that happens at the assembly site [42].

The wide range of peptides and proteins that can be expressed on the surface of filamentous phages is truly incredible. One must be aware, however, that not each and every peptide or protein can be packed into the phage capsid [42]. Especially when working with the diverse nature of cDNA libraries, the success of their display is influenced by the protein’s length, folding characteristics, stability and possible toxicity to the bacterial host [43]. The synthesis of fusion proteins, as well as the assembly of the phage virion, is determined by the bacterial physiology [42].

Accordingly, the display of cDNA libraries on phages is thus biased towards naturally secreted proteins, e.g., tumour antigens that efficiently pass through the *E. coli* translocation machinery. By contrast, the display of proteins that are less compatible with the mechanisms of phage assembly or physiological characteristics of *E. coli*, e.g., nuclear, cytoplasmic and transmembrane proteins (receptors, cell wall proteins and proteins embedded into the membrane bilayer), is hampered [18,44]. Protein fusions with sequences that do not normally cross the membrane, e.g., long stretches of hydrophobic residues that act as transmembrane stop transfer regions, also restrain the display of such cDNA gene products on phages. Equally, membrane translocation represents a drawback for large fusion proteins and thus limits the size of the DNA inserts in the phage genome. The upper limit for the size of fusion protein would be, as expected, limited by the 7 nm pIV exit pore [42]. So far, proteins up to 86 kDa have been functionally displayed [18].

Various measures have been developed to overcome these challenges described. Different signal peptides have been used to allow translocation into the periplasm [45], the folding of fusion proteins in the periplasm was improved by chaperone co-expression [46] and genetically engineered protease-deficient bacterial strains have shown a decreased rate of degradation of fusion proteins [42]. Also, employing different secretory pathways of *E. coli* enables the translocation of fusion proteins across the membrane prior to phage assembly. To date, four mechanisms have been identified in *E. coli*, with the Sec, SRP and Tat pathways requiring specific proteins for the translocation process and the fourth mechanism termed the spontaneous insertion pathway. The majority of the proteins displayed are post-translationally targeted to the membrane in an unfolded state by the Sec system, especially antibodies that require the oxidative periplasm environment for folding. To the contrary, the Tat pathway translocates folded proteins that contain metal or other cofactors and requires the reducing environment of the cytoplasm for proper folding and activity [19].

It is quite amazing how versatile, widely utilised and flexible the filamentous phage display system is, in light of all the possible constraints that the process of protein translocation and phage assembly might place on it. Beyond this, there are also technical issues that pose challenges when fusing cDNA to pIII or pVIII coat proteins. The integrity of the C-terminal end of pIII/pVIII is essential for efficient phage assembly; consequently, fusion of peptides/proteins is possible at the N-terminal end of pIII/pVIII. However, N-terminal fusion of full-length cDNA libraries is challenging due to the translational stop codon located at the 3′ end of the reverse-transcribed eukaryotic mRNA, which prevents the synthesis of fusion proteins [18,43]. Furthermore, the cDNA and leader sequence must be in the same reading frame as the pIII/pVIII gene. Since most clones contain non-functional inserts due to shifts in the reading frame or the misorientation of inserted fragments, this hinders the use of the cDNA libraries displayed on the phage [43].

The cloning strategies employed for functional phage display of cDNA libraries are represented by (a) the indirect display of peptides/proteins on the phage by means of a leucine zipper structure; (b) peptide/protein fusion to the C-terminal end of the pVI coat protein; and (c) the fragmentation of the cDNA and selection of open reading frames prior to cloning into the phage vector.

### 4.1. Indirect Fusion to pIII Coat Proteins

An elegant way to solve the problems associated with direct N-terminal fusion of cDNA fragments to pIII was developed and introduced by Crameri and Suter in 1993. They created a new phagemid vector named pJuFo. In pJuFo, they inserted the leucine zipper domains of two transcription factors, c-Jun and c-Fos. The Jun segment was linked to the N-terminus of the truncated pIII gene, and the Fos segment was linked to the N-terminus of the cDNA fragment. As the C-terminal part of the cDNA fragment is free, the stop codon does not interfere with the expression of the fusion protein. Each fusion of Jun-pIII and Fos-cDNA is separately synthesised in the bacterial cytoplasm and with an amino terminal signal sequence directed to the periplasm. There, Jun and Fos form disulfide bonds between cysteine residues, thus creating a heterodimer and indirectly enabling the linking between pIII and the displayed peptide/protein [47] (Figure 3). Phages that contain cDNA fragments cloned in the wrong reading frame or a premature stop codon will present short, unnatural peptides/proteins and will in most cases be lost during affinity selection [43].

The pJuFo system is by far the most widely exploited system for the phage display of cDNA libraries. It was shown that it enables the display of antigens in their natural conformation and retains the activity of enzymes [47,48,49]. Apart from being used for the identification of new IgE-binding molecules for different allergens [50,51,52,53,54,55], it is also utilised for the identification of new antigens in autoimmune diseases [56,57] and cancer [58]. Cloning a wide repertoire of cDNA libraries prepared from prostate cancer biopsy samples into the pJuFo phage vector and further immunoscreening with patient serum [58] have allowed for the identification of novel immunodominant cancer antigens in medicine which could be used as prognostic markers and new anti-cancer vaccine components. These antigens can be divided into tumour-associated antigens (TAAs) or tumour-specific antigens (TSAs) depending on whether they are unique to the malignant tissue (TSAs) or expressed to a varying extent on non-malignant cells (TAAs) [59,60]. With this approach, however, challenges may arise. Patients’ sera contain a heterogeneous antibody population with tumor-specific antibodies that differ in their affinities for antigens, mixed with tumour-unrelated antibodies that can interfere with the affinity selection process. Also, cDNA-encoded peptides/proteins are displayed on the phage with varying success [58]. Hence, one must note these shortcomings when evaluating the results, as the phage-displayed peptides derived from frequently represented mRNA at the time of the cDNA library construction have a selective advantage over peptides derived from rare mRNAs. Also, fewer immunogenic tumour peptides may be lost during repeated rounds of immunoselection, resulting in bias in the phage clone selection and reduced diversity in the repertoire of the selected clones [18].

It remains to be seen to what extent developments in novel phage display biopanning technologies for the enrichment and characterisation of all phage clones can further advance tumour antigen discovery studies.

### 4.2. C-Terminal Fusion to Coat Proteins pVI

Coat protein pVI is located at the same phage tip as pIII; however, it is the only protein with the N-terminus anchored into the phage capsid and the C-terminus facing outwards. Therefore, direct C-terminal fusion of the cDNA libraries is achievable, as the stop codon in the cDNA does not hinder the expression of pVI [43]. Examples of research of this kind include the selection of candidate tumour antigens through the selection of immunogenic ligands from a colon tumour cDNA repertoire with polyclonal patients’ sera. However, one shortcoming is represented by the potential selection of phage clones that display IgG transcripts as a result of the infiltration of B-cell lymphocytes into the tumour mass [44]. Hence, other studies have described antigen selection through immunoscreening phages displaying a colorectal cDNA library consisting of CRC cell lines. For the time being, the use of pVI fused cDNA libraries remains a rarity, as fusion proteins have a lower display rate than pJuFo-displayed proteins [43].

### 4.3. Direct N-Terminal Fusion to Coat Proteins pIII and pVIII

To overcome the limitations concerning the naturally occurring stop codons and untranslated regions at the 3′ end of cDNA, cDNA fragments are fragmented before cloning. Unlike the full-length cDNA sequences fundamental to studies of gene function, smaller fragments are important for identifying immunogenic epitopes in cancer [61].

According to the literature, after cloning cDNA fragments into vectors, only approximately 6% of the phagemids contained a complete open reading frame (ORF) fragment, indicating that the cDNA was correctly inserted in frame between the signal sequence gene and the pIII gene. Consequently, the constructed phage cDNA library comprised over 90% defective clones. Selection pressure favoured the propagation of phages with defective inserts or no inserts (“bald” or empty phages), which replicated more efficiently in bacterial hosts than those containing complete fragments. This phenomenon further complicated the affinity selection process by increasing the proportion of selected unwanted or unsuitable phage clones to 90–99% [62]. Thus, for effective phage display using cDNA libraries, it is essential to enrich for the correct ORFs while ensuring the successful display of the corresponding proteins or peptides [61,62,63].

The enrichment of ORF fragments allows for the cloning of cDNA fragments as fusions with the beta-lactamase gene, positioned upstream of the pIII gene. This strategy ensures that only bacterial clones with phagemids containing in-frame-inserted fragments—devoid of stop codons—can express beta-lactamase and thus survive on selective media. As a result, cDNA libraries can achieve an ORF fragment representation ranging from 54% to 100% using this enrichment method. However, this approach necessitates the subsequent removal of the beta-lactamase gene insert following positive selection employing different mechanisms [51,52,53].

## 5. General Considerations and Future Perspectives on Bacteriophages in Human Medicine

Filamentous bacteriophages serve as a valuable tool in immunotherapy for the identification of novel antigens or vaccine development [24]. They are characterised by their small size, intrinsic immunogenicity, stability over a range of pH values and resistance to hydrolytic enzymes [64]. As a consequence of their negligible tropism for mammalian cells and inability to replicate in them, coupled with their low toxicity even to the intestinal microbiome, these phages are safe for use in medicine. Phage propagation uses a prokaryotic host, which makes the production and purification of vaccines or the identification of new antigens rapid, technically simple and cost-effective at a large scale [23].

Despite the numerous advantages of phage-based immunotherapeutic agents, the path to their use in human prophylaxis and therapy remains long and uncertain, with numerous unknowns yet to be explored. Notwithstanding the current understanding of the interactions between phages, bacteria and the human immune system [65], further research is required to elucidate the effects of the additives in phage vaccines and the impact on individual health of the immune responses to phages. Further large-scale clinical trials are needed to establish their safety in humans, as are studies of the stability of the peptides displayed on these phages.

## 6. Conclusions

In conclusion, phage display technology enables the discovery of tumor-specific antigens by enabling high-throughput screening of cDNA libraries, holding promise for advancing cancer treatment. Despite the inherent challenges in phage display systems, such as limitations in the display of certain proteins and translational inefficiencies, innovative solutions and strategic improvements continue to expand their utility. By addressing these limitations and leveraging its unique advantages, phage display technology can significantly contribute to the development of effective cancer vaccines and targeted therapies, paving the way for future breakthroughs in cancer treatment.

## Figures and Tables

**Figure 1 viruses-16-01855-f001:**
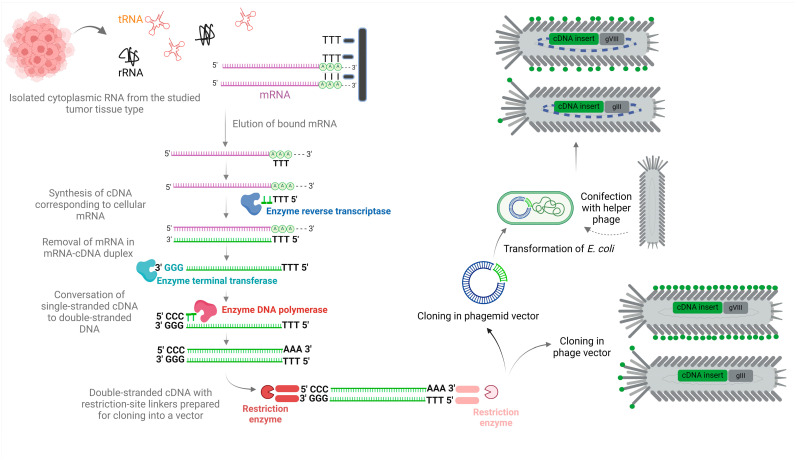
Schematic representation of the construction of cDNA libraries and their display on filamentous bacteriophages.

**Figure 2 viruses-16-01855-f002:**
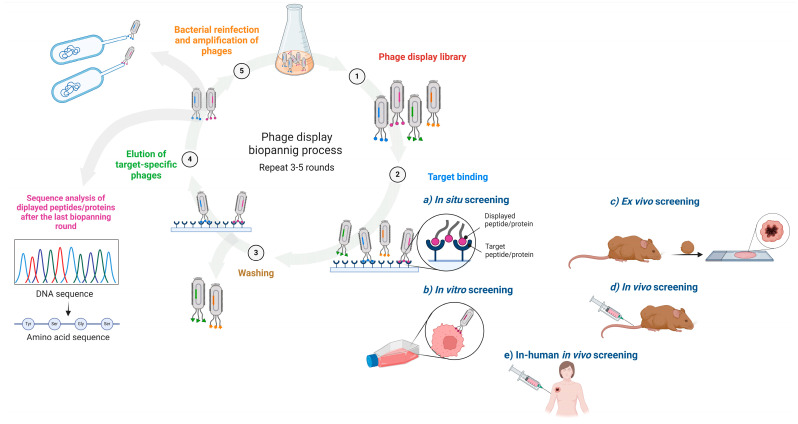
Schematic of an affinity-driven process in which phage-displayed libraries are screened against a variety of targets and target-specific phage-displayed (poly)peptides (e.g., novel cancer ligands) are subsequently identified.

**Figure 3 viruses-16-01855-f003:**
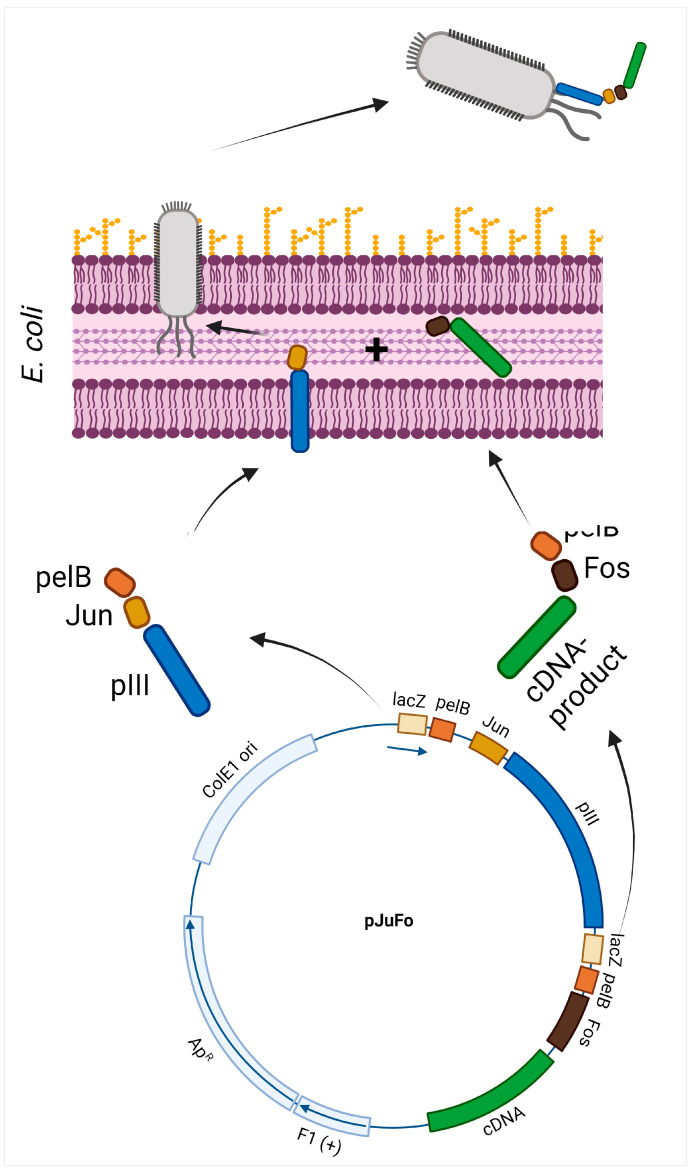
Indirect fusion to pIII coat proteins. Adapted from [47].

## Data Availability

The data presented in this study are available on request from the corresponding author.

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
