# Peer review of "Preparation of Phage Display cDNA Libraries for Identifying Immunogenic Tumor Antigens: Challenges in Functional cDNA Presentation and Approaches to Overcoming Them"

_viruses, 2024, doi:10.3390/v16121855_

Round 1
Reviewer 1 Report
Comments and Suggestions for Authors
The review entitled “Phage display of cDNA libraries and its role in the identification of new potential immunogenic tumor antigens for cancer immunotherapy” aims to describe the tumor antigens discovered in the development and advances in cancer immunotherapy; focusing on recent developments in the method of antigen cDNA libraries and Phage display. Unfortunately, the aim of the review was not met.
According to the title and aims, it would be necessary to include some sections describing tumor target peptides identified by this technology and whether they are used in the diagnosis or treatment of cancer.
The manuscript is clear and is presented in a well-structured manner to describe the construction of a cDNA library, different bioselection methods and without a description of its use in tumor targets. However, they made an excellent description of the Phage display technique using a cDNA library.
I suggest changing the title and the objective and that this manuscript could be published in a journal specialized in scientific methods.
The references cited in this review are adequate for the description of the Phage display technique. However, most of the citations are not recent (last 5 years).
The figures are adequate to describe this technology, however, they are not cited in the text.
The conclusion of the review performed is missing.
Note some errors in the text: Please change E. coli to italics (lines 238, 240, 455).
Author Response
I am most grateful to you for taking the time to review this manuscript. For your convenience, I have included the detailed responses in the attachment, and the corresponding revisions/corrections are highlighted in yellow in the re-submitted files.

Reviewer 2 Report
Comments and Suggestions for Authors
The comments are as follows:
1. In line 85, the term "Bacteriophages or short phages" is inaccurately used. It would be beneficial to provide more precise terminology.
2. In line 84, the title of the section is "Bacteriophages and Phage Display Technology," yet the content primarily focuses on filamentous phage display technology. I recommend revising the title to reflect this focus more accurately. Additionally, this subsection should include a broader overview of bacteriophages, specifically emphasizing filamentous phages.
3. Writing in line 185 requires improvement; the logical flow is unclear. Incorporating connective words would help to clarify relationships and indicate progression between ideas.
4. In lines 205-221, the first paragraph discusses displayed fusion proteins, while the subsequent paragraph addresses the packaging mechanism of phage proteins. The content should be reorganized to improve clarity and ensure logical coherence.
5. The title refers to tumor antigen screening, yet the last paragraph of the Introduction states that the article reviews the applications of these antigens in tumor immuno-therapy. Moreover, given the limited literature on tumor antigen screening, it would be beneficial to include recent research advancements in this area.
6. Throughout the manuscript, images are not properly cited in the text. It is important to reference them to appropriate locations within the manuscript. If any images are sourced from previously published literature, the corresponding references should be included.
Author Response

(The authors gave the same response as above.)

Reviewer 3 Report
Comments and Suggestions for Authors
Comments:
The review manuscript "Phage display of cDNA libraries and its role in the identification of new potential immunogenic tumour antigens for cancer immunotherapy" by Brišar et al. provides a comprehensive overview of the discovery of tumor antigens through phage display technology. It highlights the construction of cDNA libraries, their subsequent display on bacteriophages, and the application of various biopanning techniques. This article provides a concise and high-level overview of the research strategies and future directions in the field. The illustrations maintain a high level of aesthetic appeal and clarity. I believe this review will serve as a valuable guide for future discoveries of new tumor antigens using phage display technology. For specific readers, this article will be very helpful for quickly understanding the field.
Comments:
1. I recommend that the authors redraw Figure 3 to match the style of Figures 1 and 2, to achieve a cohesive aesthetic throughout the figures.
Author Response
Thank you for your helpful comment. As suggested, we have appropriately redraw Figure 3 to match the style of Figures 1 and 2, to achieve a cohesive aesthetic throughout the figures.